Corrected: Publisher correction

# Probing the strongly driven spin-boson model in a superconducting quantum circuit

L. Magazzù[1], P. Forn-Díaz[2,3,4,5], R. Belyansky[2,6], J.-L. Orgiazzi[2,4,6], M.A. Yurtalan[2,4,6], M.R. Otto [2,3,4], A. Lupascu[2,3,4], C.M. Wilson[2,6] & M. Grifoni[7]

Quantum two-level systems interacting with the surroundings are ubiquitous in nature. The interaction suppresses quantum coherence and forces the system towards a steady state. Such dissipative processes are captured by the paradigmatic spin-boson model, describing a two-state particle, the "spin", interacting with an environment formed by harmonic oscillators. A fundamental question to date is to what extent intense coherent driving impacts a strongly dissipative system. Here we investigate experimentally and theoretically a superconducting qubit strongly coupled to an electromagnetic environment and subjected to a coherent drive. This setup realizes the driven Ohmic spin-boson model. We show that the drive reinforces environmental suppression of quantum coherence, and that a coherent-to-incoherent transition can be achieved by tuning the drive amplitude. An out-of-equilibrium detailed balance relation is demonstrated. These results advance fundamental understanding of open quantum systems and bear potential for the design of entangled light-matter states.

[1] Institute of Physics, University of Augsburg, Universitätsstraße 1, D-86135 Augsburg, Germany. [2] Institute for Quantum Computing, University of Waterloo, Waterloo N2L 3G1, Canada. [3] Department of Physics and Astronomy, University of Waterloo, Waterloo N2L 3G1, Canada. [4] Waterloo Institute for Nanotechnology, University of Waterloo, Waterloo N2L 3G1, Canada. [5] Barcelona Supercomputing Center (BSC), C/Jordi Girona 29, 08034 Barcelona, Spain. [6] Department of Electrical and Computer Engineering, University of Waterloo, Waterloo N2L 3G1, Canada. [7] Institute for Theoretical Physics, University of Regensburg, 93040 Regensburg, Germany. These authors contributed equally: L. Magazzù, P. Forn-Díaz. Correspondence and requests for materials should be addressed to A.L. (email: adrian.lupascu@uwaterloo.ca) or to C.M.W. (email: chris.wilson@uwaterloo.ca) or to M.G. (email: milena.grifoni@ur.de)

The spin-boson model has been prominent for several decades in the study of open quantum systems[1, 2]. It describes a two-state quantum system (spin), interacting with its environment. The latter is modeled as a set of harmonic oscillators (bosons) constituting a so-called heat bath. The dynamical regimes of the spin-boson model at a given finite temperature are essentially dictated by the coupling to the environment and by the low-frequency behavior of the bath spectrum. In the strong coupling regime, this model provides an accurate representation of a variety of physical and chemical situations of broad interest, including incoherent tunneling of bistable defects in metals[3] and amorphous systems[4], macroscopic quantum tunneling in superconducting circuits[5], or electron and proton transfer in solvent environments[6]. Moreover, the spin-boson model is relevant in describing exciton transport in biological complexes[7, 8]. The weak coupling regime characterizes situations where preserving quantum coherence is crucial, such as in quantum computing, whereas strong coupling can give rise to novel entangled states of system and reservoir, for example, to polaron or Kondo clouds[2].

In the Ohmic spin-boson model, the environment has a linear spectrum at low frequencies which leads to various remarkable phenomena, such as bath-induced localization or a coherent-to-incoherent transition even at zero temperature for large enough coupling strengths[1].

Recently, a new experimental setup was implemented[9] which realizes the Ohmic spin-boson model with an environmental coupling tunable from weak to ultrastrong[10]. This particular implementation is formed from a superconducting flux qubit coupled to a transmission line, which play the role of the two-state system and environment, respectively. The tunability of the interaction allows one to test the key predictions of the spin-boson model. In[11], a qubit ultrastrongly coupled to a single oscillator mode was demonstrated.

In this article, we study the spin-boson setup from ref.[9] under strong driving, which adds a new dimension of exploration for a spin-boson system[12]. Previous experiments studying strongly driven systems have reported remarkable effects, such as the formation of dressed states[13–15], Landau-Zener interference[16, 17], amplitude spectroscopy[18], and the observation of Floquet states[19]. However, these experimental reports were restricted to weak or moderate coupling to the environment. Here, we combine intense driving and diverse dissipation strengths in a superconducting qubit circuit, with the aim of tracing out the dynamical phase diagram of a driven spin-boson system in coupling regimes ranging form weak to ultrastrong.

## Results

**Relation between experimental and theoretical observables.** A schematic representation of the experimental setup is shown in Fig. 1a. The two-state system is a flux qubit, a superconducting circuit consisting of a loop interrupted by four Josephson junctions[20]. The bosonic environment is formed from electromagnetic modes in the superconducting transmission line coupled to the qubit. The qubit is pumped by a strong continuous-wave drive applied through the transmission line. Both the amplitude and the frequency of the drive can be changed over a broad range. The driven system is studied spectroscopically by additionally applying a weak probe field. The measured transmission $\mathcal{T}$ at the probe frequency $\omega_p$ gives direct access to the linear response function associated to the weak probe signal, the so-called linear susceptibility $\chi$ via the relation

$$\mathcal{T}(\omega_p) = 1 - i\mathcal{N}\hbar\omega_p\chi(\omega_p), \qquad (1)$$

where $\mathcal{N}$ is a coupling constant (see Methods). According to

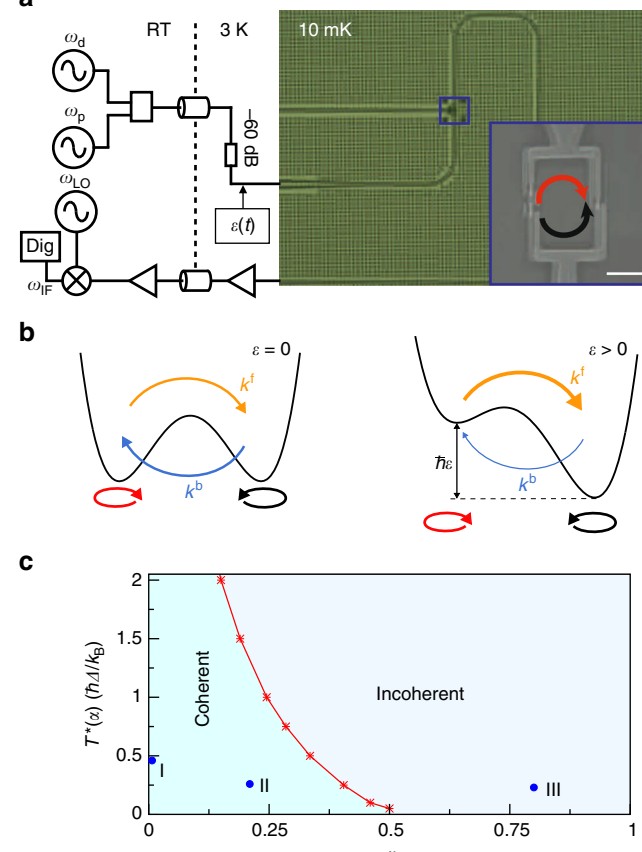

**Fig. 1** Experimental setup and phase diagram of the symmetric spin-boson model. **a** Measurement, driving circuit schematic and optical micrograph of a device similar to the ones used in the experiment. A coplanar waveguide running across the chip plays the role of the bath coupled to the qubit. The inset is a scanning electron micrograph showing the qubit attached to the line. The scale bar is 2 μm. Here and in panel **b** the red (black) arrow indicates clockwise (anticlockwise) circulating persistent currents. **b** Schematics of the double-well potential associated to the flux threading the qubit. In the absence of external driving sources the potential is symmetric and the forward and backward tunneling rates $k^{f/b}$ are equal. In the presence of a positive bias asymmetry $\varepsilon$, forward tunneling dominates over backward tunneling. **c** Dependence of the temperature $T^*(\alpha)$ for the crossover from the coherent to the incoherent tunneling regime on the coupling $\alpha$. The red curve interpolates numerical results (asterisks) obtained within the nonperturbative NIBA. The dots labeled I, II, and III mark the positions in parameter space of the three devices used in this work

Kubo's linear response theory[21], $\chi(\omega)$ carries information about the dispersive and absorptive properties of the qubit in the absence of the probe, and in turn, as discussed below, about the dynamical phases of the driven spin-boson system. By measuring the transmission also when the drive is switched off, we get a reference for the effects of a coherent drive on quantum coherence and localization properties.

**Phase diagram of the undriven spin-boson model.** We first introduce the spin-boson model and its dynamics without driving. Historically, the Ohmic spin-boson model was first studied in the context of the tunneling of a quantum particle in a double-well potential[1]. At low temperatures the dynamics are effectively restricted to the Hilbert space spanned by the states $|L\rangle$ and $|R\rangle$,

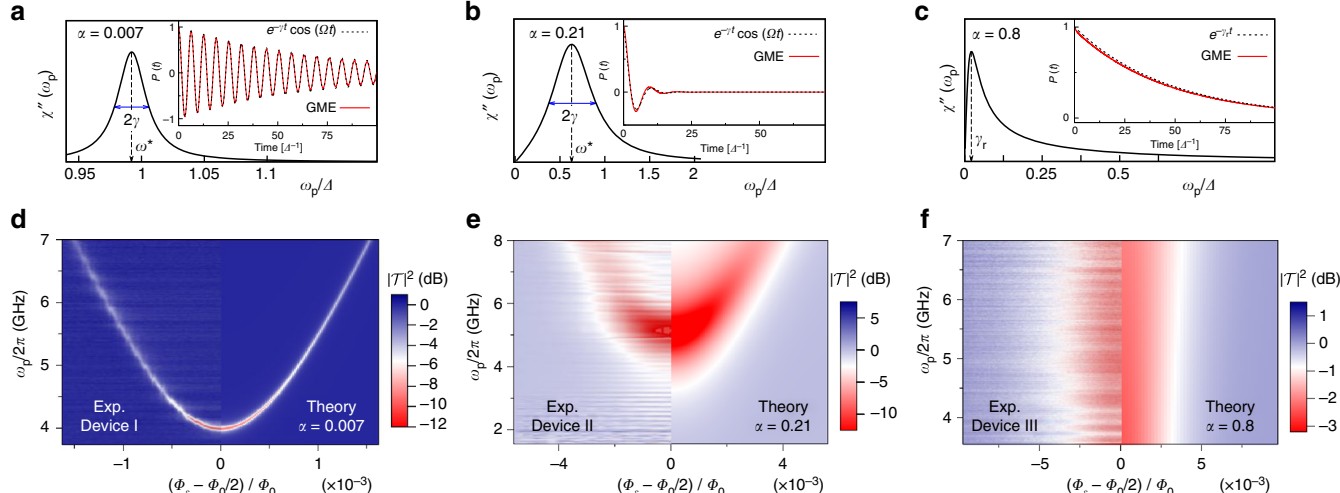

**Fig. 2** Spin-boson dynamics and spectra at different coupling strengths in the absence of the drive. **a–c** Frequency dependence of the imaginary part $\chi''(\omega_p)$ of the linear susceptibility (a.u.) and time evolution of the population difference $P(t)$ (insets) for the three selected combinations of coupling and temperature shown in Fig. 1c. The position $\omega^\star$ and FWHM $2\gamma$ of the linear susceptibility peak in the coherent regimes ($\alpha = 0.007$, $\alpha = 0.21$) provide a direct measure of the renormalized qubit frequency $\Omega = \sqrt{(\omega^\star)^2 - \gamma^2}$. In the incoherent regime ($\alpha = 0.8$), the peak position yields the relaxation rate $\gamma_r$. **d–f** Experimental transmission spectra of three flux qubit devices with different coupling junctions are compared with spectra calculated within the NIBA. The characteristic hyperbolic spectrum of the flux qubit is evident in **d** and recognizable in **e**. Its disappearance in **f** indicates the transition to the incoherent regime. At $\Phi_\varepsilon = \Phi_0/2$ the spin-boson system is unbiased, which is the situation of panels **a–c**

localized in the left and right well, respectively (see Fig. 1b). Transitions between the two localized configurations are possible due to quantum-mechanical tunneling and are recorded in the time evolution of the population difference $P(t) \equiv \langle \sigma_z(t) \rangle = P_R(t) - P_L(t)$ of the two localized eigenstates. The coordinate associated with the double-well potential need not be geometrical, but it can represent other continuous variables. For the superconducting flux qubit used in our experiment, this is the magnetic flux $\Phi$ in the loop. The eigenstates $|L\rangle$ and $|R\rangle$ of the flux operator are related to the currents circulating clockwise/anticlockwise in the superconducting loop[20] (see red/black arrows in Fig. 1a, b). In this basis, the qubit Hamiltonian is

$$H_{qb}(t) = -\frac{\hbar}{2}[\Delta \sigma_x + \varepsilon(t)\sigma_z], \qquad (2)$$

where $\sigma_i$ are the Pauli matrices. The parameter $\Delta$ accounts for interwell tunneling and $\hbar\varepsilon(t)$ is the difference in energy between the two wells, which is controllable. The electromagnetic field in the transmission line can be described as a continuously distributed set of propagating modes with a distribution in frequency given by the spectral density

$$G(\omega) = 2\alpha\omega e^{-\omega/\omega_c}, \qquad (3)$$

corresponding to Ohmic damping with the dimensionless coupling strength $\alpha$ and high frequency cutoff $\omega_c$.

Theoretical work on the spin-boson model has primarily focused on the temporal dynamics of the spin. Quite generally, independent of the initial state of the qubit and the form of the bath spectral density, energy exchange with the environment is responsible for equilibration of the qubit with the bath on a time scale given by the relaxation rate $\gamma_r$. Furthermore, quantum fluctuations and energy exchange yield dephasing with rate $\gamma$. In the Ohmic spin-boson model, low frequency environmental modes also lead to a strong renormalization of the bare qubit tunneling splitting $\Delta$. The renormalized qubit frequency $\Omega$ depends on the bath temperature and coupling strength $\alpha$, and is always reduced with respect to $\Delta$. This leads to three distinct dynamical regimes. Two of them, occurring for $\alpha < 1$, are depicted

in Fig. 1c for the symmetric spin-boson model shown in the left drawing in Fig. 1b. The coherent regime corresponds to $\Omega > \gamma$. This occurs for $\alpha < 1/2$ and a temperature $T < T^\star(\alpha)$. In this regime, for a spin initially localized in the right well ($P(0) = 1$), the qubit displays damped coherent oscillations of frequency $\Omega$, specifically, $P(t) = \exp(-\gamma t)\cos(\Omega t)$ (see insets of Fig. 2a, b). At the crossover temperature, the renormalized frequency $\Omega$ vanishes (see Methods and Eq. (26)). The incoherent regime corresponds to $\alpha < 1/2$ and $T > T^\star(\alpha)$ or $1/2 < \alpha < 1$. The dynamics are characterized by incoherent tunneling transitions with rates $k^{f/b}$ defined in Section III of the Methods section (Fig. 2b). Correspondingly, we have $P(t) = e^{-\gamma_r t}$, where $\gamma_r = k^f + k^b$ (see inset in Fig. 2c). In the third regime, corresponding to $\alpha > 1$, localization occurs. Here, the backward and forward rates are renormalized to zero by the low-frequency bath modes. As shown in Fig. 1c, in the Ohmic spin-boson model, the dynamics becomes fully incoherent above $\alpha = 0.5$ for any value of the temperature. As the coupling approaches this value, any perturbative approach in the coupling fails to describe the physics of the system. Consistently with ref. [9], we refer to the coupling regimes $\alpha > 0.5$ as ultrastrong. Primary scope of this work is to understand how the dynamical phase diagram in Fig. 1c is modified by a periodic modulation of the detuning. This is a formidable task, since the spin-boson problem with time-periodic detuning cannot be solved analytically in the whole parameter space. Exact solutions exist for the particular value $\alpha = 1/2$[22]. Recently, an analytical solution was suggested for the case of a spin-boson system with time-periodic tunneling amplitude[23].

**Linear susceptibility of the driven spin-boson model.** To carry out our spectroscopic analysis, we describe the bias between the potential wells in our experimental setup by means of the time-dependent function

$$\varepsilon(t) = \varepsilon_0 + \varepsilon_p \cos(\omega_p t) + \varepsilon_d \cos(\omega_d t). \qquad (4)$$

Here, the static component $\varepsilon_0$ is related to the externally applied flux $\Phi_\varepsilon$ by $\varepsilon_0 \propto (\Phi_\varepsilon - \Phi_0/2)$, with $\Phi_0$ the magnetic flux quantum. The remaining contributions account for the probe (p), with

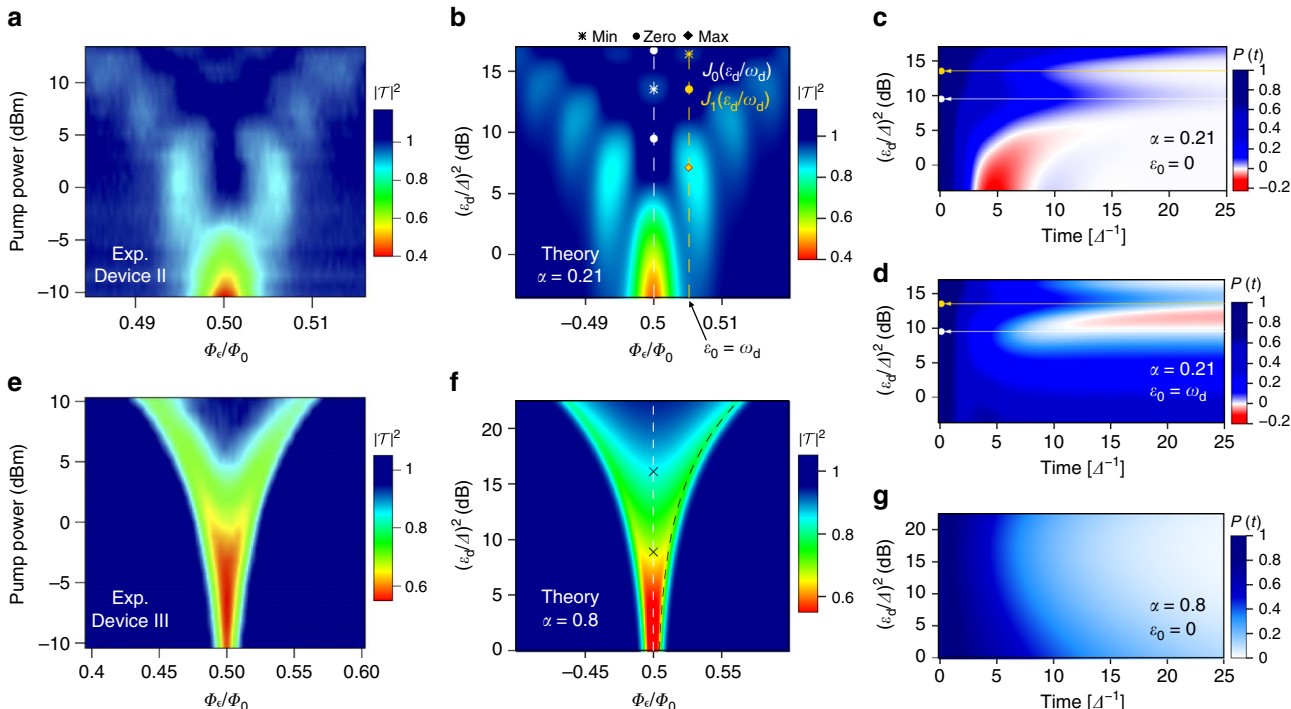

**Fig. 3** Spectral response and dynamics of the driven spin-boson system. **a**, **b** Observed and calculated transmission at the probe frequency for the moderately coupled Device II ($\alpha = 0.21$) as function of static bias and pump strength. A clear structure of multi-photon resonances appears. The dashed lines indicate cuts at fixed bias where the dynamics in panels **c**, **d** are calculated. **c**,**d** Predicted dynamics of $P(t)$ with $P(0) = 1$. **e**, **f** Observed and calculated spectrum of the ultrastrongly coupled Device III (with $\alpha = 0.8$). The spectrum is smoothed, as compared to panels **a**, **b**, indicating fully incoherent dynamics. The black dashed line in **f** corresponds to the condition $\varepsilon_{\text{eff}} = 2k_{\text{B}}T/\hbar$ for the effective nonequilibrium bias (see Eq. (7)) and the symbols "×" mark the first two zeroes of $J_0(\varepsilon_d\tau_{\text{env}})$. **g** Time evolution of $P(t)$ calculated at the symmetry point, $\varepsilon_0 = 0$, in the same range of pump strengths as in panel **f**

amplitude $\varepsilon_{\text{p}}$ and frequency $\omega_{\text{p}}$, and the drive (d), with amplitude $\varepsilon_{\text{d}}$ and frequency $\omega_{\text{d}}$. For details, see the Methods section. The central quantity in this work is the linear susceptibility $\chi(\omega_{\text{p}})$, which describes the qubit's response at the probe frequency $\omega_{\text{p}}$, see Eq. (1). The susceptibility measures deviations of the asymptotic population difference, $P^{\text{as}}(t)$, from its value $P_0$ in the absence of the weak probe according to[24]

$$P^{\text{as}}(t) = P_0 + \hbar\varepsilon_{\text{p}}\left[\chi(\omega_{\text{p}})e^{i\omega_{\text{p}}t} + \chi(-\omega_{\text{p}})e^{-i\omega_{\text{p}}t}\right]. \quad (5)$$

In this work, the dynamical quantity $P(t)$, and in turn the susceptibility $\chi(\omega_{\text{p}})$, have been calculated within the so-called noninteracting-blip approximation (NIBA). This approximation yields a generalized master equation for $P(t)$ with kernels that are nonperturbative in $\alpha$. It becomes exact at large temperatures and/or coupling strengths[2]. Under the assumption that $\omega_{\text{d}}$ is large compared to the (renormalized) frequency scales of the spin-boson particle, closed expressions for the transient evolution of $P(t)$, as well as for the linear susceptibility of the driven spin-boson system, can be obtained (details in the Methods section).

**Characterizing the dynamical regimes of the undriven devices**. We first demonstrate in Fig. 2a–c the connection between the imaginary part, $\chi''(\omega_{\text{p}})$, of the susceptibility and $P(t)$ for the symmetric spin-boson model in the presence of the probe only ($\varepsilon_0 = \varepsilon_{\text{d}} = 0$). We choose three distinct values of the coupling, namely $\alpha = 0.007, 0.21$, situated in the coherent regime, and $\alpha = 0.8$ in the incoherent regime (see the three dots indicated in Fig. 1c). In the coherent regime, $\chi''(\omega_{\text{p}})$ has a peak at $\omega^* = (\Omega^2 + \gamma^2)^{1/2}$ with full width at half maximum (FWHM) given by $2\gamma$. In the incoherent regime, the peak is located near zero frequency, at the value of the relaxation rate $\gamma_{\text{r}}$. According to Eq. (1), a

maximum in $\chi''(\omega_{\text{p}})$ corresponds to a minimum in the transmission $\mathcal{T}(\omega_{\text{p}})$. By recording the evolution of the transmission as a function of $\omega_{\text{p}}$ and of another external parameter, e.g., the static asymmetry $\varepsilon_0$, various dynamical regimes can be identified.

The theoretically calculated transmission is presented in Fig. 2d–f as a function of the applied static bias $\varepsilon_0$ for the three values of $\alpha$ discussed above. As expected, the qubit dispersion relation can be traced back in the highly coherent and underdamped regimes corresponding to $\alpha = 0.007$ and $\alpha = 0.21$, respectively. In the overdamped regime, with $\alpha = 0.8$, the transmission is nearly independent of $\omega_{\text{p}}$. Finally, comparison with the measured transmission for three distinct tunable devices, named I, II, and III in the following, allows us to position the three devices as shown in the phase diagram in Fig. 1c. Temperature, cutoff frequency, renormalized splitting $\Omega$, and conversion factor $\mathcal{N}$ are estimated from the experiments. Deviations in the choice of these parameters can yield variations in the estimate of the coupling strength $\alpha$. The close agreement between the calculated and measured qubit spectra gives a strong evidence that Device III, with an estimated coupling $\alpha = 0.8 \pm 0.1$ (see the Supplementary Note 7), is in the nonperturbative ultrastrong coupling regime, buttressing the conclusion of ref. [9, 25]. In a recent work[26] a polaron approach, which is equivalent to the NIBA[2], has been used to provide approximate expressions for the response of an undriven qubit coupled to a transmission line.

**Spectroscopy of the driven spin-boson model**. Let us now turn to the impact of a strong coherent drive on a spin-boson particle in the intermediate and ultrastrong coupling regimes captured by devices II and III, respectively. The experimental spectra in Fig. 3a, e show the probe transmission as a function of flux bias $\varepsilon_0$

and drive power ($\propto \varepsilon_{\mathrm{d}}^2$) for these devices. Probe and drive frequencies are respectively set to $\omega_{\mathrm{p}}/2\pi = 5.2$ GHz and $\omega_{\mathrm{d}}/2\pi = 9$ GHz for Device II. For Device III we choose $\omega_{\mathrm{p}}/2\pi = 4$ GHz and $\omega_{\mathrm{d}}/2\pi = 3$ GHz. For Device II, the probe is on-resonance with the undriven qubit at the symmetry point. For Device III, the qualitative features of the driven spectra are largely insensitive to the choice of $\omega_{\mathrm{p}}$ and $\omega_{\mathrm{d}}$. The theoretical predictions, shown in Fig. 3b, f, agree well with the experimental observations. Similar to the pump-only case, striking differences are observed in the transmission of the two devices. Let us start discussing Device II. Minima in the transmission are clearly seen in Fig. 3a, b whenever the static bias matches a multiple of the pump frequency, $\varepsilon_0 = n\omega_{\mathrm{d}}$, as indicated by the vertical lines drawn in Fig. 3b for $n = 0$, 1. Furthermore, the observed pattern with fixed bias at the $n$-th resonance results from a modulation by a prefactor proportional to $J_n(\varepsilon_{\mathrm{d}}/\omega_{\mathrm{d}})$, where $J_n$ is a Bessel function of the first kind. For example, the qubit response at the symmetry point is suppressed in correspondence with the first zero of the Bessel function $J_0(\varepsilon_{\mathrm{d}}/\omega_{\mathrm{d}})$ (indicated by a circle), where the incoming probing field is fully transmitted. At larger power, as the zero order Bessel function increases again, the transmission diminishes. Similar patterns have already been reported in driven qubit devices in the highly coherent regime[14, 17]. Those results can be interpreted as a signature of entangled light-matter states known as dressed-states[13, 27, 28]. Near the multiphoton resonance, $\varepsilon_0 = n\omega_{\mathrm{d}}$, two of these dressed states form an effective two-level system with dressed tunneling splitting $\Delta_n = \Delta J_n(\varepsilon_{\mathrm{d}}/\omega_{\mathrm{d}})$. Near a zero of the $n$-th Bessel function, tunneling is strongly suppressed and hence transmission is maximal. This phenomenon has been dubbed coherent destruction of tunneling in the literature[29]. Dissipation modifies this simple coherent picture, as demonstrated for Device III in Fig. 3e, f where no Bessel pattern is present and a smooth "V-shaped" transmission is observed instead.

## Discussion

To understand to what extent dissipation modifies the dressed state picture, we have studied the transient dynamics of the population difference $P(t)$ in the presence of drive only ($\varepsilon_{\mathrm{p}} = 0$). As discussed in the Methods, $P(t)$ is governed by a generalized master equation featuring the two nonequilibrium kernels $\mathcal{K}^{+/-}(t)$, which in the absence of probe field, are symmetric/antisymmetric in the static bias $\varepsilon_0$. In Laplace space, by solving the pole equation $\lambda + K^+(\lambda) = 0$, where $K^+(\lambda) = \int_0^\infty \exp(-\lambda t)\mathcal{K}^+(t)$, the phase diagram of the driven spin-boson particle can in principle be found along the lines discussed in the Methods. The kernel $K^+(\lambda)$ can be expressed as the sum $K^{\mathrm{f}}(\lambda) + K^{\mathrm{b}}(\lambda)$ of the nonequilibrium forward and backward kernels

$$K^{\mathrm{f/b}}(\lambda) = \frac{\Delta^2}{2}\int_0^\infty \mathrm{d}t\, e^{-Q'(t)-\lambda t}J_0[d(t)]\cos[Q''(t)\mp\varepsilon_0 t], \quad (6)$$

with $d(t) = 2\varepsilon_{\mathrm{d}}\omega_{\mathrm{d}}^{-1}\sin(\omega_{\mathrm{d}}t/2)$. The correlation function $Q(t) = Q'(t) + iQ''(t)$ describes the environmental influence and its explicit form is discussed in the Supplementary Note 1 and in Eqs. (15) and (16) of the Methods. For the present discussion, it is enough to observe that in the long-time limit $t \gg \tau_{\mathrm{env}}$, where $\tau_{\mathrm{env}} = (2\pi\alpha k_{\mathrm{B}}T/\hbar)^{-1}$, the real part of $Q(t)$ assumes the form $Q'(t) \sim t/\tau_{\mathrm{env}} + \mathrm{const.}$ appropriate to white noise. Thus, $\tau_{\mathrm{env}}$ yields an estimate of the memory time of the kernels entering Eq. (6). The impact of the drive is encapsulated in the time-dependent argument of the Bessel function of first kind $J_0$. Depending on whether $\omega_{\mathrm{d}}\tau_{\mathrm{env}} \gtrsim 1$ (slow relaxation) or $\omega_{\mathrm{d}}\tau_{\mathrm{env}} \lesssim 1$ (fast relaxation), two distinct regimes corresponding to devices II and III are encountered, respectively.

Let us focus on the first case, explored in Fig. 3a, b. In this regime, one full cycle of the drive field is possible before environmental effects induce a loss of coherence. Thus, we expect that coherent absorption and emission processes from the drive field take place during a cycle. An expansion of the Bessel function in Eq. (6) in a Fourier series, $J_0[d(t)] = \sum_n J_n^2(\varepsilon_{\mathrm{d}}/\omega_{\mathrm{d}})\exp(in\omega_{\mathrm{d}})$, shows that the channel with $n\omega_{\mathrm{d}} = \varepsilon_0$, dominates the series[12], and hence an effective two-level description with renormalized tunneling splitting $\Delta_n$ applies. A solution of the pole equation in this approximation yields a renormalization of the crossover temperature $T^\star(\alpha) \to T^\star(\alpha)[J_n(\varepsilon_{\mathrm{d}}/\omega_{\mathrm{d}})]^{1/(1-\alpha)}$. Because $J_n < 1$, the pump field always yields a reduction of quantum coherence. Near the zeros of $J_n$, quantum coherence is fully suppressed and an incoherent decay is expected. This behavior is seen in Fig. 3c, d, where we show the simulated time evolution of $P(t)$ as a function of pump power at $\varepsilon_0 = 0$ and $\varepsilon_0 = \omega_{\mathrm{d}}$, respectively. The color map of $P(t)$ displays coherent oscillations at low to moderate pump amplitudes, where $J_0(\varepsilon_{\mathrm{d}}/\omega_{\mathrm{d}})$ is still of order one. However, a full suppression of quantum coherence occurs near the first zero of $J_0$, highlighted by a solid white circle. We notice that the almost complete standstill predicted to occur at the zeros of $J_0$ for a dissipation-free, symmetric two-level particle[29], is destroyed by environmental relaxation processes, albeit on a very slow time scale. A similar suppression of coherence, together with a very slow incoherent decay, is observed at the first resonance, $\varepsilon_0 = \omega_{\mathrm{d}}$, shown in Fig. 3d, in correspondence with the first zero of $J_1$. Independently of the initial preparation, the steady state population acquires the value $P_0 = (K^{\mathrm{f}} - K^{\mathrm{b}})/(K^{\mathrm{f}} + K^{\mathrm{b}})$, where $K^{\mathrm{f/b}} = K^{\mathrm{f/b}}(\lambda = 0)$ are the nonequilibrium backward and forward rates. For the symmetric case shown in Fig. 3c, the backward and forward rates are equal and hence $P_0 = 0$. A genuine nonequilibrium behavior is observed in Fig. 3d in the region between the first zeros of $J_0$ and $J_1$, where the steady state qubit population $P_0 < 0$, corresponding to a larger population of the left state despite $\varepsilon_0 > 0$. This phenomenon originates from the effective detailed balance relation

$$K^{\mathrm{f}} = K^{\mathrm{b}}e^{\hbar\varepsilon_{\mathrm{eff}}/k_{\mathrm{B}}T} \quad (7)$$

between the nonequilibrium backward and forward rates $K^{\mathrm{f/b}}$. This equation implicitly defines the effective asymmetry $\varepsilon_{\mathrm{eff}}$. Only in the absence of the drive does $\varepsilon_{\mathrm{eff}}$ coincide with the static bias $\varepsilon_0$. We note that the use of an external coherent drive to tune the direction of long-range electron chemical reactions via a drive-induced effective bias was originally proposed in refs. [30, 31].

Let us turn to the explanation of the results for Device III displayed in Fig. 3e–g, where $\omega_{\mathrm{d}}\tau_{\mathrm{env}} \ll 1$ applies. In this regime the approximate result

$$\chi(\omega_{\mathrm{p}}) = \frac{1}{4k_{\mathrm{B}}T}\frac{\partial\varepsilon_{\mathrm{eff}}/\partial\varepsilon_0}{\cosh^2(\hbar\varepsilon_{\mathrm{eff}}/2k_{\mathrm{B}}T)}\frac{\gamma_{\mathrm{d}}}{\gamma_{\mathrm{d}} + i\omega_{\mathrm{p}}} \quad (8)$$

can be obtained from the exact expression Eq. (19) of the Methods section. This form is associated to the incoherent dynamics of the spin boson particle with nonequilibrium relaxation rate $\gamma_{\mathrm{d}} \equiv K^{\mathrm{f}} + K^{\mathrm{b}}$. At the symmetry point we have $\varepsilon_{\mathrm{eff}} = \varepsilon_0 = 0$, with $\lim_{\varepsilon_0\to 0}\partial\varepsilon_{\mathrm{eff}}/\partial\varepsilon_0 \neq 0$. Correspondingly, the susceptibility $\chi''(\omega_{\mathrm{p}})$ has a peak at $\omega_{\mathrm{p}} = \gamma_{\mathrm{d}}$. An expansion in the small parameter $\omega_{\mathrm{d}}\tau_{\mathrm{env}}$ yields $J_0[d(t)] \approx J_0(\varepsilon_{\mathrm{d}}t)$ and hence a relaxation rate $\gamma_{\mathrm{d}}$ which is independent of the driving frequency $\omega_{\mathrm{d}}$, consistent with the experimental observation that the spectra depend weakly on $\omega_{\mathrm{d}}$. The dependence on the pump amplitude $\varepsilon_{\mathrm{d}}$ remains, as clearly seen in Fig. 3e–g where the transmission at the symmetry point smoothly increases for increasing drive amplitude. The transmission is almost complete for drive powers above the value $(\varepsilon_{\mathrm{d}}/\Delta)^2 \simeq 16$ dB roughly corresponding to the second

**Table 1 Parameters used for simulations**

| Figures 2 and 3 | Device I | Device II | Device III |
|---|---|---|---|
| $\omega_c/2\pi$ (GHz) | 65 | 65 | 65 |
| $I_{pers}$ (nA) | 600 | 280 | 250 |
| $\alpha$ | 0.007 [fit] | 0.21 [fit] | 0.8[a] |
| $\Delta/2\pi$ (GHz) | 4.04 [fit] | 7.23 [fit] | 8.0[a] |
| **Figure 2** | **Device I** | **Device II** | **Device III** |
| $T$ (mK) | 90 | 90 | 90 |
| $\mathcal{N}$ (estimated) | 0.03 [fit] (0.02) | 1.1 [fit] (0.5) | 8.0[a] (5–10) |
| **Figure 3** | | **Device II** | **Device III** |
| $T$ (mK) | | 175[a] | 90[a] |
| $\mathcal{N}$ | | 1.1[a] | 16.0[a] |
| $\omega_p/2\pi$ (GHz) | | 5.2 | 4.0 |
| $\omega_d/2\pi$ (GHz) | | 9.0 | 3.0 |

[a]Value yielding qualitative agreement with the experiment, see Supplementary Note 7

zero of $J_0(\varepsilon_d\tau_{env})$ (see Fig. 3f, where the black crosses highlight the first two zeroes). Regarding the transmission at finite static bias, we expect that no thermally assisted excitation is possible when $\hbar\varepsilon_{eff} \gg k_BT$; correspondingly the susceptibility vanishes, as accounted by the term $\cosh^{-2}(\hbar\varepsilon_{eff}/2k_BT)$ in Eq. (8). This behavior is clearly seen in Fig. 3f, where the black dashed line corresponds to the condition $\hbar\varepsilon_{eff} = 2k_BT$. Below the dashed line the effective bias is larger than the temperature and the signal is fully transmitted.

In conclusion, we have experimentally and theoretically explored the paradigmatic driven spin-boson model in the underdamped and ultrastrong dynamical regimes. Quantum coherence is generally reduced or even destroyed by a drive field in a way which can be tuned by sweeping the drive amplitude and frequency. The control of the dynamics is possible for a generic Ohmic spin-boson particle, independently of its microscopic details. Localization and even population inversion can be attained by properly tuning the parameters of the coherent drive. Our results might find application in various physical, chemical, and quantum biology realizations of the driven spin-boson model.

## Methods

**Experimental fabrication and measurement setup.** Devices were fabricated according to the procedure explained in ref. [9]. Our setup was designed in such a way that the reservoir (the photons in the transmission line) can still be considered in equilibrium despite the strong pumping applied to the qubit. The response of the photons depends on the intensity of the drive and on the coupling mechanisms. In our experiment, the degrees of freedom of the bath are very weakly coupled to the drive, compared to the qubit. Hence, even though the qubit is strongly driven, the bath is not. To be more quantitative, the most sensitive component of our bath is the 50 Ohm input of our amplifier. From its data sheet, the amplifier starts to become nonlinear for an input power of −12 dBm (its 1 dB compression point), which is many orders of magnitude higher than what our pump power is. The other components of our bath, which would be microwave attenuators (resistors), are linear up to energies a few orders of magnitude higher. From the theoretical point of view, we expect that the transmission of the fully-driven spin-boson model would differ qualitatively from the one of the system-driven spin-boson model considered in this work. No trivial mapping exists between the two models. The very good agreement between theoretical predictions and the experiment validate our conclusion that merely the system is driven.

**Relation between theoretical and experimental observables.** The flux operator in the qubit basis is identified with $\hat{\Phi} = f\sigma_z$. The proportionality constant $f$ is a fitting parameter which, for low couplings, is estimated to be $f = MI_{pers}$, with $M$ the qubit-line mutual inductance and $I_{pers}$ the persistent current in the super-conducting loop. This estimate provides values (see Table 1) which are not far from those obtained from fit to data for devices I and II and from qualitative analysis for Device III. The externally applied tunable flux $\Phi_\varepsilon$ is related to the static bias by $\hbar\varepsilon_0 = 2I_{pers}(\Phi_\varepsilon - \Phi_0/2)$, with $\Phi_0$ the magnetic flux quantum. The probe input voltage is connected to the angular frequency $\varepsilon_p$ yielding the theoretical probe amplitude, see Eq. (4), through $V_p^{in}(t) = f_Z\varepsilon_p\cos(\omega_pt)$, where the proportionality constant is $f_Z =$

$\hbar Z/f$ and $Z$ is the line impedance. It follows that the constant $\mathcal{N}$ in Eq. (1) is given by the ratio $f/f_Z$.

**Parameters used in the simulation.** The parameters used in the numerical simulations shown in Figs. 2 and 3 are provided in Table 1. Coupling $\alpha$, bare tunneling frequency $\Delta$, and proportionality constant $\mathcal{N}$ are determined by fit to data of $|\mathcal{T}|^2$ vs. $\omega_p$ performed for the nondriven devices I and II at the symmetry point $\Phi_\varepsilon = \Phi_0/2$ (see Fig. 2d, e). Such fits along with their accuracy are shown in Supplementary Fig. 4. In Fig. 2, the measured value of 90 mK is used for the temperature. Temperature values used in Fig. 3 account for a possibly higher effective temperature introduced by the drive at the qubit position. Specifically, for Device II, in the presence of the pump drive, a better qualitative agreement between simulated and experimental transmission is obtained by assuming a higher temperature. As the qualitative features of the simulated transmission for Device III, operating at ultrastrong coupling, are weakly sensitive to variations of the temperature, we used the same value of temperature for the pump-probe and the probe-only cases.

**Driven spin-boson dynamics within the NIBA.** The spin-boson model describes the coupling of a two-level quantum system to a bath of harmonic oscillators[32]. By assuming a coupling which linearly depends on the coordinates of the oscillators, one arrives at the famous spin-boson Hamiltonian

$$H(t) = H_{qb}(t) - \frac{\hbar}{2}\sigma_z\sum_i c_i\left(a_i^\dagger + a_i\right) + \sum_i \hbar\omega_i a_i^\dagger a_i, \quad (9)$$

where $a_i$, $a_i^\dagger$ are bosonic annihilation and creation operators and the coefficients $c_i$ are the amplitude of the interaction strength of the two-level system with mode $i$. The bosonic heat bath is fully characterized by the spectral function $G(\omega) = \sum_i c_i^2\delta(\omega - \omega_i)$. For Ohmic damping, $G(\omega) \propto \omega$, as assumed in Eq. (3).

The Ohmic spin-boson problem owes its popularity to its ubiquity and to the variety of parameter regimes it encompasses as the temperature $T$ and the coupling strength $\alpha$ are varied. We refer the readers to ref. [2] for an exhaustive treatment. The dynamical properties of a driven spin-boson system in the strongly damped and in the incoherent regimes, are well described within the so-called noninteracting-blip approximation (NIBA). Furthermore, the NIBA captures well the dynamics of a symmetric ($\varepsilon_0 = 0$) spin-boson system in the whole parameter regime. The NIBA approximation provides a generalized master equation (GME) for the evolution of the population difference $P(t)$ with rates in second order in the bare tunneling splitting $\Delta$ but nonperturbative in $\alpha$. Accounting for the presence of time dependent fields, the GME explicitly reads

$$\dot{P}(t) = \int_{t_0}^t dt'\left[\mathcal{K}^-(t, t') - \mathcal{K}^+(t, t')P(t')\right]. \quad (10)$$

The NIBA kernels $\mathcal{K}^\pm$, averaged over a pump period, are given by

$$\mathcal{K}^+(t, t') = h^+(t - t')\cos\left[\zeta(t, t')\right], \quad (11)$$

$$\mathcal{K}^-(t, t') = h^-(t - t')\sin\left[\zeta(t, t')\right], \quad (12)$$

with

$$h^+(t) = \Delta^2 e^{-Q'(t)}\cos[Q''(t)]J_0\left[\frac{2\varepsilon_d}{\omega_d}\sin\left(\frac{\omega_dt}{2}\right)\right], \quad (13)$$

$$h^-(t) = \Delta^2 e^{-Q'(t)}\sin[Q''(t)]J_0\left[\frac{2\varepsilon_d}{\omega_d}\sin\left(\frac{\omega_dt}{2}\right)\right]. \quad (14)$$

The function $Q(t) = Q'(t) + iQ''(t)$ is the environmental correlation function. For the Ohmic spectral density function $G(\omega) = 2\alpha\omega\exp(-\omega/\omega_c)$, $\alpha$ being the dimensionless coupling strength and $\omega_c$ a high frequency cutoff, and in the scaling limit $\hbar\omega_c \gg \beta^{-1} = k_BT$, these functions have an explicit form[2]

$$Q'(t) = 2\alpha\ln\left[\sqrt{1 + \omega_c^2t^2}\frac{\sinh(\pi t/\hbar\beta)}{\pi t/\hbar\beta}\right], \quad (15)$$

$$Q''(t) = 2\alpha\arctan(\omega_ct). \quad (16)$$

The above formulas are accurate in all coupling regimes, provided that the cutoff frequency is large with respect to the other frequency scales involved. In the long-time limit ($t/\beta\hbar \gg 1$) the real part of $Q(t)$ assumes the form $Q'(t) \sim t/\tau_{env} +$ const., where $\tau_{env} = (2\pi\alpha k_BT/\hbar)^{-1}$. Thus the latter quantity determines the memory time of the kernels $\mathcal{K}^\pm$ in Eqs. (11) and (12).

The dynamical phase entering the kernels reads

$$\zeta(t, t') = (t - t')\varepsilon_0 + \frac{\varepsilon_p}{\omega_p}\left\{\sin(\omega_p t) - \sin[\omega_p(t')]\right\}. \tag{17}$$

Note that in the absence of the probe field, $\varepsilon_p = 0$, the pump-averaged kernels depend only on the difference $t - t'$, i.e., $\mathcal{K}^\pm(t, t') = \mathcal{K}^\pm(t - t')$, as in the static case. The latter is then recovered by additionally setting $\varepsilon_d = 0$. On the other hand, the probe-only setup is described by Eq. (10) upon setting $\varepsilon_d = 0$ in Eqs. (13) and (14). The dynamics shown in the insets of Fig. 2a–c are based on the numerical solution of the GME (10) for $\varepsilon(t) = 0$, whereas in the time evolution of $P(t)$ vs. pump power shown in panels c, d, and g of Fig. 3, only the probe field is set to zero.

**The linear susceptibility**. The linear susceptibility is related to the asymptotic probability difference by

$$P^{as}(t) = P_0 + \hbar\varepsilon_p\left[\chi(\omega_p)e^{i\omega_p t} + \chi(-\omega_p)e^{-i\omega_p t}\right], \tag{18}$$

where, in the NIBA, $P_0$ reduces to the equilibrium value $P_{eq} = \tanh(\hbar\varepsilon_0/2k_B T)$ in the absence of pump driving. The transmission $\mathcal{T}(\omega_p)$ and the susceptibility $\chi(\omega_p)$ shown in the theoretical plots of Figs. 2 and 3 are calculated by means of the exact NIBA expression[12]

$$P_0 = \frac{K^-(0)}{K^+(0)}, \quad \chi(\omega_p) = \frac{H^+(\omega_p) - H^-(\omega_p)P_0}{i\omega_p + K^+(i\omega_p)}, \tag{19}$$

with superscripts $\pm$ denoting symmetric/antisymmetric functions of $\varepsilon_0$. For our pump-probe case we find

$$H^+(\omega_p) = \frac{1}{\hbar\omega_p}\int_0^\infty dt\, e^{-i\omega_p t/2}\sin\left(\frac{\omega_p t}{2}\right)h^-(t)\cos(\varepsilon_0 t), \tag{20}$$

$$H^-(\omega_p) = \frac{-1}{\hbar\omega_p}\int_0^\infty dt\, e^{-i\omega_p t/2}\sin\left(\frac{\omega_p t}{2}\right)h^+(t)\sin(\varepsilon_0 t), \tag{21}$$

$$K^+(\lambda) = \int_0^\infty dt\, e^{-\lambda t}h^+(t)\cos(\varepsilon_0 t), \tag{22}$$

$$K^-(\lambda) = \int_0^\infty dt\, e^{-\lambda t}h^-(t)\sin(\varepsilon_0 t). \tag{23}$$

Here $K^\pm(\lambda) = \int_0^\infty d\tau\, e^{-\lambda\tau}\mathcal{K}^\pm(\tau)$ are the Laplace transforms of the pump-averaged kernels in Eqs. (11) and (12) with $\varepsilon_p = 0$. The kernels $K^\pm(\lambda)$ are related to the forward and backward rates $K^{f/b}(\lambda)$, introduced in Eq. (6), by $K^\pm = K^f \pm K^b$. Also, the incoherent rates for the static case are defined as $k^{f/b} = K^{f/b}(\lambda = 0, \varepsilon_d = 0)$. For devices I and II, in the absence of pump driving, we analytically evaluated the integrals in Eqs. (20), (21), (22), and (23) and used the resulting expressions in the susceptibility $\chi$, Eq. (19), to perform fits to the data. In the limit $\omega_p\tau_{env} \ll 1$, Eq. (19) simplifies to Eq. (8) of the main text (see the Supplementary Note 4).

**Coherent-to-incoherent transition**. In the absence of probe driving, $\varepsilon_p = 0$, the population difference $P(t)$ is conveniently obtained by introducing the Laplace transform $\hat{P}(\lambda) = \int_0^\infty dt\, e^{-\lambda t}P(t)$. From Eq. (10) one finds

$$\hat{P}(\lambda) = \frac{1 + K^-(\lambda)/\lambda}{\lambda + K^+(\lambda)}. \tag{24}$$

The pole in $\lambda = 0$ determines the asymptotic value $P_0 = K^-(0)/K^+(0)$ reached at long times. The solution of the equation $\lambda + K^+(\lambda) = 0$ yields information on the transient dynamics. In the underdamped regime, complex solutions yield the renormalized tunneling frequency with associated dephasing rate. In the incoherent regime, the long-time dynamics is ruled by a single exponential decay with relaxation rate $\gamma_d \equiv K^+(\lambda = 0)$, see Eq. (22).

Let us focus exemplarily on the undriven spin-boson system at the symmetry point $\varepsilon_0 = 0$. Then, an expansion around $\lambda = 0$ yields a quadratic equation for the poles of $\hat{P}(\lambda)$[33]. In the coherent regime the roots are complex conjugated, $\lambda_{1,2} = -\gamma \pm i\Omega(T)$, while they are real in the incoherent regime (cf. insets in Fig. 2a–c). The temperature $T^*$ at which the oscillation frequency $\Omega(T)$ vanishes determines the transition between the coherent and incoherent regimes. For weak coupling one finds for example $\Omega = \Delta_r(1 - \pi\alpha\hbar\Delta_r/k_B T)$ with

$$\Delta_r = \Delta(\Delta/\omega_c)^{\alpha/(1-\alpha)}g(\alpha) \tag{25}$$

and $g(\alpha) = [\Gamma(1 - 2\alpha)\cos(\pi\alpha)]^{1/2(1-\alpha)}$. This allows the estimate $T^*(\alpha) \approx \hbar\Delta_r(k_B\alpha)^{-1}$ when $\alpha \ll 1$. For general $\alpha < 1$ it is given by

$$T^*(\alpha) \approx \frac{\hbar\Delta_r}{k_B}[\Gamma(\alpha)/\alpha\Gamma(1 - \alpha)]^{1/2(1-\alpha)}, \tag{26}$$

where $\Gamma(x)$ is the Euler Gamma function. This approximate expression matches well the numerically calculated crossover temperature shown in Fig. 1c. The coherent-incoherent transition temperature $T^*(\alpha)$ depicted there is established, for $\alpha < 0.5$, by using Eq. (19), with numerically evaluated kernels, whereas the point at $\alpha = 0.5$ is individuated by the exact result $k_B T^*(\alpha = 0.5)/\hbar\Delta = \Delta/2\omega_c^2$. Further details are found in the Supplementary Note 9.

**Data availability**. The data that support the main findings of this study are available from the corresponding author upon request.

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

## Acknowledgements

The authors acknowledge financial support by the Deutsche Forschungsgemeinschaft via SFB 631, NSERC of Canada, the Canadian Foundation for Innovation, the Ontario Ministry of Research and Innovation, Industry Canada and Canadian Microelectronics Corporation. L.M. gratefully acknowledges financial support by the Angelo Della Riccia Foundation and hospitality by the Regensburg University during the early stages of the project. P.F.-D. is supported by the Beatriu de Pinós fellowship (2016BP00303). The authors thank J.J. García-Ripoll, B. Peropadre, and P. Hänggi for fruitful discussions, and S. Chang, A. M., and C. Deng for help with device fabrication and with the measurement setups.

## Author contributions

L.M. and M.G. performed the theoretical analysis, with numerical simulations carried out by L.M. The experiments were designed and performed by P.F.-D., A.L., and C.M.W. The devices were fabricated by P.F.-D., J.-L.O., M.A.Y., and M.R.O. contributed to device design and fabrication. R.B. assisted in numerical modeling of the device. The manuscript was mainly written by M.G. with critical comments provided by all authors. The supplementary information was mainly written by L.M.. The experimental work was a collaboration between the labs led by A.L. and C.M.W.

## Additional information

**Competing interests:** The authors declare no competing interests.

