## [Peer Review File · Nature Communications]

REVIEWERS' COMMENTS:

Reviewer #1 (Remarks to the Author):

reviewer report for NCOMMS-17-32301-T

"Probing the strongly driven spin-boson model in a superconducting quantum circuit"

Looking at the previous correspondence, my feeling is that the authors did sufficiently address issues raised.

The key result of this work is the experimental implementation of the driven spin-boson model, building on the previously obtained theoretical expertise. In particular, the authors can validate the signatures of the predicted phases.

Major flaws in the argument of the manuscript are not apparent to me.

The results in the paper are based on previous experimental work of the authors for the undriven spin-boson model.

In particular, the model is one of the canonical models for open quantum systems, it arises as a cartoon

in many different contexts. Adding periodic driving to these models will lead to the driven version of the spin-boson model,

and therefore I believe that the results in this manuscript may be interesting for a large scientific community and

researchers from different physics disciplines. The way the article is written makes it accessible mainly to physicists.

In the introduction, the authors could also make some effort to connect to applications of the spin-boson model in other

fields such as quantum biology, but this is in my opinion optional.

Regarding the clarity of the methods, I think the authors could make some more efforts to explain why they locate their device III

only with qualitative agreement of the plots around $\alpha=0.8$. I think here one would either like to see a fit with error bars

(as was done for devices I and II) or, alternatively, a more detailed explanation why a similar approach has not been followed

for device III, together with estimates on numerical simulations on the qualitative agreement around $\alpha=0.8$, perhaps

adding some uncertainty region around the points in Fig. 1c.

I do not have the expertise to judge whether the experimental methods are adequately described, but from the theoretical perspective

I think that the work represents a sound contribution and would therefore be inclined to recommend publication after the suitable revisions

have been made.

minor things:

+ I would suggest the authors to add their response to reviewer #3 in the previous correspondence to the discussion of the spin

boson model in the main article (regarding the "strongness" of the coupling for any $\alpha>0$).

+ explanation regarding the +/- and f/b relation in Eq. (6) missing

+ comma missing in Eq. (17)

+ I personally would prefer it if non-running indices and also the imaginary unit are set in roman style,
but at least this should be handled consistently throughout the manuscript

Point-by-point reply to the Reviewer #1 NCOMMS-17-32301A

"In the introduction, the authors could also make some effort to connect to applications of the spin-boson model in other fields such as quantum biology, but this is in my opinion optional."

- In the Introduction, when discussing the applicability of the spin-boson model, we introduced a comment on the relevance of the model in the context of exciton transport in biological structures, referring in particular to the new Refs. [7] and [8], which were added to the bibliography. In these works the spin-boson model is used explicitly to model the transport of electronic excitations in the presence of a vibrational environment.

"Regarding the clarity of the methods, I think the authors could make some more efforts to explain why they locate their device III only with qualitative agreement of the plots around $\alpha=0.8$. I think here one would either like to see a fit with error bars (as was done for devices I and II) or, alternatively, a more detailed explanation why a similar approach has not been followed for device III, together with estimates on numerical simulations on the qualitative agreement around $\alpha=0.8$, perhaps adding some uncertainty region around the points in Fig. 1c."

- We added to the Supplementary Information a section (Supplementary Note 7) where the estimate of the coupling strength $\alpha=0.8$ and its uncertainty (± 0.1) are discussed. At the end of the Subsection 'Characterizing the dynamical regimes of the undriven devices' we state the uncertainty associated to $\alpha=0.8$ and refer to the new Supplementary Note 7.

"minor things:

+ I would suggest the authors to add their response to reviewer #3 in the previous correspondence to the discussion of the spin boson model in the main article (regarding the "strongness" of the coupling for any $\alpha>0$)...."

- At the end of Subsection 'Phase diagram of the undriven spin-boson model' we introduced a comment about the terminology used (ultrastrong coupling), on the basis of our reply to the report of Reviewer #3.

"....+ explanation regarding the +/- and f/b relation in Eq. (6) missing"

Above Eq. (6) we clarified the relation between the kernel K^+ and the forward/backward kernels $K^{f/b}$.

"+ comma missing in Eq. (17) + I personally would prefer it if non-running indices and also the imaginary unit are set in roman style, but at least this should be handled consistently throughout the manuscript"

- We updated the manuscript and the Supplement, also with respect to the notation, according to the editorial guidelines and the suggestions of the Reviewer

List of Changes in the main text

- Stylistic and formatting changes according to the guidelines and suggestions by the Editor plus minor changes regarding few typos and capitalization/hyphenation to uniform the notation
- Introduction:
Sentence about the spin-boson model in the context of Quantum Biology with the two additional references [7] and [8]
- New Figs. 1-3 with improved readability (with only minor changes in Fig. 1)
- End of Subsection 'Phase diagram of the undriven spin-boson model':
Added a discussion about the terminology used (ultrastrong coupling)
- End of the Subsection 'Characterizing the dynamical regimes of the undriven devices':
Statement of the uncertainty associated to $\alpha=0.8$ and reference to the new Supplementary Note 7
- Above Eq. (6):
Clarified the relation between the kernel K^+ and the forward/backward kernels K^f/b .
- References:
New bibliographical entries [7] and [8]

List of changes in the Supplementary Information

- Stylistic and formatting changes according to the guidelines and suggestions by the Editor plus minor changes regarding few typos and capitalization/hyphenation to uniform the notation
- New, improved Supplementary Figures. 2-8. Additional curve in Supplementary Figure 3
- New Supplementary Note 7 discussing the parameters regimes of Device III
- Phase diagram with coherent/incoherent transition removed. We now refer to the identical diagram in Fig. 1c of the main text.
- Updated Supplementary Reference [4]